# “Your Life Turns Upside Down”: A Qualitative Study of the Experiences of Parents with Children Diagnosed with Phelan-McDermid Syndrome

**DOI:** 10.3390/children10010073

**Published:** 2022-12-30

**Authors:** Cristina García-Bravo, Domingo Palacios-Ceña, Elisabet Huertas-Hoyas, Jorge Pérez-Corrales, Sergio Serrada-Tejeda, Marta Pérez-de-Heredia-Torres, Javier Gueita-Rodríguez, Rosa Mª Martínez-Piédrola

**Affiliations:** 1Department of Physical Therapy, Occupational Therapy, Physical Medicine and Rehabilitation, Research Group in Evaluation and Assessment of Capacity, Functionality and Disability, Universidad Rey Juan Carlos, 28922 Alcorcón, Spain; 2Department of Physical Therapy, Occupational Therapy, Physical Medicine and Rehabilitation, Research Group of Humanities and Qualitative Research in Health Science, Universidad Rey Juan Carlos, 28922 Alcorcón, Spain

**Keywords:** Phelan-Mcdermid Syndrome, Telomeric 22q13 Monosomy Syndrome, rare diseases, parents, qualitative research

## Abstract

(1) Background: Parents of children with rare diseases experience great uncertainty and employ different strategies to care for their children and cope with the disease. The purpose of the present study was to describe the perspective of parents with children with Phelan McDermid Syndrome (PMS). (2) Methods: A non-probabilistic purposeful sampling was used to perform this qualitative descriptive study. Thirty-two parents with children with PMS were interviewed. In-depth interviews and research field notes were analyzed using an inductive thematic analysis. (3) Results: Four themes emerged from the data. “Understanding and accepting the disease” described how parents experienced their child’s diagnosis and the lack of information. The second theme, called “Living day by day”, highlighted the daily difficulties faced when caring for a child with PMS. The third theme, “Expectations versus reality”, was based on the parents’ expectations of parenthood and the reality they face. Expectations for the future are also included. Finally, “Pain and happiness” describes how parents alternate feelings of distress and suffering but also joy with what they learn from these experiences. (4) Conclusions: Health professionals can use these results to support parents.

## 1. Introduction

Phelan-McDermid Syndrome (PMS), is caused by the loss of one or more nucleotides in the DNA sequence near the end segment of chromosome 22 that disrupts the SHANK3 gene and the synthesis of the SHANK3 protein [1]. The alteration of this gene and of the protein causes psychomotor delay, intellectual disability, the absence or severe delay of speech and the presence of autistic traits [2,3]. In addition, it may be accompanied by nonspecific dysmorphic features, behavioral problems, seizures/epilepsy, neuropsychiatric decompensations (bipolar disorder or catatonia) and sleep or cardiac abnormalities [4,5].

PMS is thought to be underdiagnosed and its real incidence is unknown due to the difficulty of diagnosis because it is not detected by conventional chromosomal analysis. It is estimated that there are about 2000 people in the world with PMS, and it is considered a rare disease [3,5]. Currently, there is no treatment for PMS. The therapeutic effort is focused on support for families, the treatment of comorbidities such as behavioral problems, sleep disturbances, sensory processing, seizures/epilepsy or respiratory problems [6,7] and the management of limitations and associated disability [8,9].

The experience of parents with children with PMS has never been described by a qualitative study. Accordingly, the purpose of this study was to explore and describe the experience of parents with children diagnosed with PMS regarding how they integrate the disease into their daily life, expectations regarding the disease and treatment, and the challenges and rewards involved in the daily care of children with PMS.

## 2. Materials and Methods

### 2.1. Study Design

A qualitative descriptive study was performed [10,11] according to the Standards for Reporting Qualitative Research (SRQR) [12].

### 2.2. Participants, Context, and Sampling Strategies

The study included parents who were attending an association for families with PMS. We adopted as inclusion criteria the following: parents and/or legal guardians whose children were diagnosed with PMS by the pediatrician and/or the neurologist (confirmed by genetic diagnosis) at the time of the study; and any variation of PMS (deletion or mutation) could be included.

The sampling strategy used was the purposive sampling, considering the relevance to the research question, not the clinical representativeness [10]. A minimum sample size of 30 participants was chosen as, according to the Turner-Bowker et al. proposal [13], 99.3% of concepts, themes, and contents emerge with around 30 interviews [13].

### 2.3. Ethical Considerations

Ethical approval was provided by the University Rey Juan Carlos Ethical Committee (code: 0810202017820). Prior to data collection, participants were asked for permission and informed consent was obtained. The data obtained were anonymized and were not available to anyone outside the research team.

### 2.4. Data Collection

Data were collected between November 2020 and February 2021. Data were collected through in-depth interviews guided by a semi-structured question guide (Table 1). In addition, researcher field notes were kept [14]. The investigators used prompts or probes: (a) to encourage patients to elaborate further (Can you tell me more about that?), (b) to encourage participants to keep talking (Have you experienced the same thing again?), (c) to solve doubts (paraphrase some participant’s sentence), and (d) to demonstrated that they have the researcher’s full attention (That’s very interesting, please tell me more).

The Microsoft Teams platform (https://www.microsoft.com/es-es/microsoft-teams/log-in (accessed on 2 October 2022) was used to perform interviews using private video call. CGB, RMMP and DPC conducted a total of 32 interviews. A total of 3205 min of interviews were audio and video-recorded (average 100.16 ± 18.2 min), with the previous oral consent of each participant. Additionally, 32 field notes were collected by the researchers during the semi-structured interviews. Field notes provide a rich source of information about the participants’ behaviors during their experience description, and researcher’s reflections concerning methodological aspects of the data collection [14].

### 2.5. Data Analysis

An inductive thematic analysis was performed using the full verbatim transcription of each of the interviews, and the researchers’ field notes [10,14]. The analysis consisted of identifying the most descriptive content to obtain codes, and subsequently reduce and identify the most common meaningful groups (categories). Groups of meaningful units were formed (i.e., similar points or content that enabled the emergence of the topics that described the study participants’ experience) [10,14]. This process was performed separately on the interviews and the researchers’ field notes. Additionally, double, and independent coding was performed by CGB, DPC. Thereafter, the results of the analysis were obtained after discussion, comparison and refinement consensus of both investigators [10,14]. Investigators organized and shared their coding process was organized and shared by the Excel program.

### 2.6. Rigor

Trustworthiness was controlled by the four techniques described in Table 2 [10,15].

## 3. Results

A total of thirty-two parents with a mean age of 43.38 years old (SD:6.85) were interviewed, and 23 of them were women. The mean age of children with PMS was 11.66 years (SD:9.79) and the mean age of the child at diagnosis was 7.04 years (SD:10.04) with a mean evolution of the diagnosis of 4.62 years (SD:4.88). The clinical and demographic features of each participant are shown in Table 3. The participants included in the present study were one of the parents, except in cases E1–E2, E6–E7, and E15–E16, where both spouses from the same family agreed to participate.

Four specific themes emerged: (a) understanding and accepting the disease; (b) living day by day; (c) expectations versus reality; and (d) suffering and happiness. Below, some of the participant’s narratives are directly cited to illustrated the emerging themes.

### 3.1. Theme 1: Understanding and Accepting the Disease

This theme describes the parents’ experiences of the appearance of the first symptoms, together with uncertainty, and the lack of information. The parents described how they detected changes in their children that did not fit in with their expectations of development or with what they had been told. In some cases, they recounted how they did not want to notice the symptoms: “Suddenly there are things that don’t add up and as a new parent you tell yourself that it’s nothing... you don’t want to see it... but there comes a time when you say to yourself ‘there’s something wrong here’.” (P24).

The parents acknowledged feeling unprepared for PMS, as something that does not enter into their life plans, meaning that the possibility of any problem of this type was never considered: “It’s something they never prepare you for... When you become pregnant, at no time does this, this dependence, cross your mind and your world crumbles. You think of what could have been and isn’t, and it breaks your soul…” (P2). This experience is considered “very hard” at first. Other parents emphasized the lack of information, with feelings of uncertainty and difficulty in facing the unknown, especially regarding the future care of their child: “The worst thing for me is the uncertainty that I don’t know how far [my son] will be able to go or what will become of him.” (P7).

Parents described mood changes, and how they stopped living their lives (e.g., professionally) to prioritize the care of their children: “You give up everything for your child. And in the end, you don’t move forward. Yes, it has meant sacrifices of things you wanted to have done. For example, my professional life.” (P3).

In addition, many compare PMS to Down Syndrome, thus PMS does not have physical characteristics that can be identified with difficulties or having a disease: “… this is not Down Syndrome… it is not something that people can see physically. Nor is it something that is known so that people can understand it.” (P21).

### 3.2. Theme 2: Living Day by Day

Parents experience day-to-day life as a constant adaptation where nothing can be planned, because taking care of a child takes priority over any plan: “Before, we used to plan everything and, in the end, we had a hard time when we couldn’t do it... you learn to be more flexible and if something has to be changed, we adapt and we do it.” (P26). The main focus is on the child with PMS, with the rest of the family taking a back seat: “Just picture it. Everything revolves around him, everything. Our priority is not us, what’s important is him.” (P22).

Parents described how all resources (time and money) are allocated to the care and management of children with PMS, which leads to difficulties in reconciling work and maintaining relationships with their circle of friends, who gradually stop counting on them: “… you don’t go out as much... with so much therapy you don’t have time to go out and the little time you do have is spent taking care of her. In the end even your friends stop calling you.” (P13). They emphasized that they continually had to face challenges, and the main struggle is having to explain themselves to people, especially in relation to the child’s disruptive behavior, which others fail to understand: “…When a child misbehaves in a public place, the first thing people think about is the lack of attention from the parents. They don’t think there’s anything wrong with them... your normal day-to-day life is a continuous struggle... it’s exhausting....” (P23).

Parents reported feeling like they are always on alert as the child needs constant attention and this causes them constant stress and tension: “In the end it affects you, I, for example, am a very light sleeper. I hear any noise and I think it’s her. I’m always alert, and I don’t rest.” (P3). Many parents described their day-to-day life as non-stop, “like a locomotive” that cannot stop because everything goes so fast and they don’t have time to think or reflect: “You can’t allow yourself to think, nor philosophize too much because if you do you would collapse... everything just comes down on you, there’s no time for anything.” (P4).

### 3.3. Theme 3: Expectations versus Reality

This theme describes the parents’ expectations of building a family, of being parents, including their understanding of normality, versus the real future that unfolds when you have a child with PMS. The diagnosis of PMS meant a rupture in their life project and in forming a family: “When we got married we decided to have a child to build our family, our project, and suddenly, the syndrome is there and all that project goes down the drain.” (P23). Many parents recounted the expectations and life plans they had when they became pregnant and before the diagnosis of PMS. They described their plans for their children such as the school they were going to attend, how they were going to play with them, their professional future, and how the diagnosis of PMS erased and faded all their plans. The parents acknowledged feelings of suffering when they compared themselves to other children and parents: “I had always wanted to have a son to play soccer with him, to take him to watch soccer. Now I see my friends going with their children playing soccer and it’s impossible not to compare yourself with them. That’s what hurts me the most.” (P7). Some fathers and mothers said that they live a different normality compared to other parents, however, it is their normality because they have not experienced anything else: “For me it’s a different normality because it’s our daily life, we haven’t experienced anything else. It may be different, but it’s our normality…” (P11).

In relation to the future, parents are met with a reality that does not allow them to have many expectations. They expressed the anguish, frustration and uncertainty that appears when they do not know how far they will be able to go and explained how their expectations and hopes diminish day by day: “…we don’t have much expectation [of a cure], we just hope we don’t make it worse...” (P22). Most parents reported that they avoided thinking about the future because it is painful, and uncertain. They preferred to live life day by day and plan only for the short to medium term in order to be happy: “… We try not to think too much about the future. We learned to live in the present, because if I think about the future, you can’t live, what will happen to him when he is older?” (P6). Moreover, regarding their expectations about their own life they emphasized their uncertainties regarding the future, of their own health, and whether they will have the strength it takes to care for their child: “I am strong now, but as you age you get weaker, and my son has more strength, and with the crises when he gets agitated... the day will come when I won’t be able to take care of him. That worries me a lot.” (P8). In addition, they spoke of the possibility of their children needing care when they (the parents) are away. Most parents spoke of how siblings will have to take a leading role in caregiving: “I think we are a team; she didn’t choose it either [another sibling] It’s her brother, she’s going to get involved, she’s going to be helpful and in the future, she’s going to take care of him.” (P25).

### 3.4. Theme 4: Suffering and Happiness

During their daily lives, parents alternate between moments of suffering and moments of great happiness. The parents spoke of very painful and sad moments, crying on a daily basis. They spoke of “a pain that you carry inside” and that you must endure it in order to understand it. Parents described a feeling of guilt for not having been able to prevent the onset of the syndrome in their children, for thinking about what their lives could have been like if their child died, for not knowing enough to help their children, for not being able to give them all the resources and help they need, for not understanding them and sometimes not being able to communicate, or for not recognizing the symptoms: “… I have gone through a lot of guilt, but mostly because of thinking ugly things like how my life would change if he had not passed the acute phase. That’s the hardest thing I’ve ever thought, it’s very overpowering” (P6). However, they reported feeling joy and happiness because they are better people thanks to their children. Their children give them unconditional love, have changed their perspective on life and have taught them not to judge people: “Once my other son said something that really stuck with me. He told me that thanks to his brother we were all better people, and he’ s right. We always think about the bad things, but there are good things too.” (P13).

## 4. Discussion

### 4.1. Understanding and Accepting the Disease

PMS is characterized by a wide variety of clinical presentations including challenging behaviors, sleep disorders, autism spectrum disorder (ASD), or intellectual disability [1,16]. Previous studies of parents of children with rare diseases (RD) showed that is a challenging and exhausting process. In addition, the studies described the need to understand the specific needs of the children, the difficulties in caring for them and the concern for their behavior [17,18]. Our participants experienced the disease as a constant challenge and an extremely hard process for which they are unprepared, since they face an unknown syndrome with a great variety of symptoms, and a lack of treatment. The lack of knowledge, the lack of information about RD, and the difficulty in obtaining truthful information, are aspects highlighted by parents in previous studies [17,19]. Further information is needed to cope with day-to-day life and symptom management. Moreover, previous studies have reported the lack of information that health professionals have about RD [20,21].

Previous studies on RD [19,21,22,23] showed how parents are tired and fatigued by the daily struggle of caring for their children. In addition, they point out the great impact that the disease has on their lives and that of the family, due to the psychological and economic overload. Our participants show similar results, focusing all aspects of their lives on the care of their children.

The parents interviewed remarked that they often have to provide explanations for their child’s disruptive behavior. Previous studies on the perspective of parents of children with RD showed a lack of understanding of the pathology itself and of the children’s disruptive behaviors [24,25,26]. The presence of physical signs of the disease may help to understand the disruptive behaviors. Children with PMS may present with nonspecific dysmorphic features such as large fingers, long eyelashes, and large ears, although these features are not characteristic of the syndrome [1,26]. However, in Down Syndrome, where the presence of physical traits is more marked, parents report the same lack of understanding. They are confronted with comments from people on a daily basis as a result of the child’s behavior [27]. Regardless of whether or not physical features are present, Currie and Szabo [24] report that in the face of signs of RD and when complications become visible to others, they feel a stigma from the rest of society that takes an emotional toll on them.

### 4.2. Living Day by Day

Our results coincide with previous studies on the environment and daily care of children with RD. These studies showed how parents must adapt a multitude of tasks due to limitations in ADLs, become socially isolated, change their routines due to a lack of time, increase economic expenses, focus the family’s efforts on caring for the child and present difficulties in accessing medications [11,17,22,25]. Children with PMS have a high level of dependency and require personal and environmental support for daily functioning [28,29].

The presence of behavioral problems and/or autistic traits in PMS appears most frequently between childhood and adolescence [5,7,30]. Nayeri et al. [31] reflect the difficult situations experienced by parents due to the misunderstanding that people display towards the behaviors of their children with a RD. Thus, the generalized experience of the parents of children with a diagnosis of RD, Down Syndrome, or ASD, shows a generalized lack of understanding of others regarding their children’s disruptive behaviors [24,25,26,27]. As a result, they report a constant struggle to avoid stigma and a lack of understanding by the public. It involves mental and physical overexertion and exhaustion, as well as stress [24,32]. Bolbocean et al. [33] and Vogels et al. [34] state that the families of people with PMS are at high risk of mental health problems such as depression, stress and/or anxiety associated with the caregiving role.

One of the aspects highlighted by participants is the challenge of knowing all the information related to the disease in order to provide the best possible care for their children. Our parents recognize that, in addition to learning to be parents of children with disabilities, they must learn to be experts on everything. Regardless of the type of RD, parents become experts on the pathology and the care of the children and all their needs, which leads to a considerable increase in roles [24,26,35].

### 4.3. Expectations versus Reality

The difficulty in recognizing the first symptoms and not accepting the existence of the disease is an experience shared by parents of children with RD [35,36]. In children with severe disabilities, parents experience a change in their role, accompanied by uncertainty, followed by understanding [17,36]. In addition, previous studies [31,37] reported how the construction of meaning about parenthood, the cultural and social expectations linked to the meaning of what a family is, influences the expectations that parents consider “normal” for their children and for themselves. Rare diseases shatter all the meanings and expectations that parents had constructed about their child, the family, and their future [31,37]. Previous studies [17,38] showed the uncertainty and fear that parents have about the future of their children when they grow up. A frequent concern identified in previous studies is the death and/or dependency (associated with old age) of the parents and their ability to continue caring for their children [17,38]. Our participants also expressed concern for the future and the need to leave everything prepared (resources, responsible person) so that their child’s needs are covered, especially economic needs.

Sometimes, parents compare their children with other healthy children, which is especially painful for them. However, parents report that, after doing so, they regret these comparisons, especially in the early stages of RD, as it is a very hard process for them and causes great suffering [24].

### 4.4. Suffering and Happiness

Caring for your child is a very hard and painful process for parents, yet, at the same time, it brings them great happiness. Parents with children with a RD described a great psychological burden due to the unknown disease itself, and great difficulty handling daily care. In addition, they experience stress, uncertainty, constant fatigue and/or limitations in several personal areas [24,39]. Another aspect highlighted by parents of children with PMS is the constant struggle they deal in all areas (health, education, institutions, etc.). This is a common characteristic reflected in other qualitative studies on RD or children with disabilities [22,23,39]. Nonetheless, parents of children with a RD describe that the feelings they experience are like a roller coaster, they may cry while also feeling happy and proud of their children [17,40]. Moreover, previous studies have shown the appearance of feelings of guilt and suffering in the day-to-day life of the parents of children with a RD. Awareness of the present and future limitations of children with a RD causes a situation that they define as difficult and painful for the parents [24,31,36,41,42]. However, living with a child with a RD shows them the positive and important things in life [24].

Comparing our results with other RD, a change in the perspective of parents with children with a RD is described to the extent that their understanding of the disease improves [18], they become aware of their children’s needs and care requirements [24], they learn to manage the multiple symptoms of the RD, and their expectations are adapted to the specific reality of each family [24,43]. Nonetheless, the future is a sensitive issue, because parents express limited confidence that a drug or curative treatment for the disease will be available for their children [43,44].

The authors of the present study believe that a key difference between parents with children with PMS and/or RD, and parents with children with severe health problems, is that families with PMS and/or RD are more vulnerable because: (a) PMS has no known curative treatment [7,45]; (b) there are problems related to its diagnosis that result in a diagnostic delay [46]; and (c) there are not enough resources in the health system for the study of PMD and/or RD because they are a minority [2,45,46].

This study has limitations concerning generalizability, as results of a qualitative study cannot be extrapolated to all parents and/or legal guardians who have children with PMS. However, the impact of PMS on a parent’s routine and their management of the disease, reported by our participants, may help and guide professionals to understand and to assist parents. Moreover, the themes emerged may help describe and explore other phenomena, such as the impact of rehabilitation therapies on children, and the learning and self-care processes experienced by parents. Another limitation is that the questions asked to the parents have not been selected in accordance with the child’s age (e.g., birth-3 years, 4–8 years, etc.), nor have they been asked at specific moments in time, such as immediately after diagnosis. This could enable inquiries into the parents’ perspective on aspects such as the increase or decrease of certain behaviors or other symptoms at different developmental stages. In addition, future studies could include questions that would be useful for other parents with children with other RD. For example, they could include: (a) What do you wish you’d been told at the time you found out about the diagnosis? At any other earlier stages? (b) Has more time since the diagnosis changed your expectations or outlook on the disease? (c) What are the best resources you have encountered? Moreover, we only registered the presence of deletion and SHANK3 gene mutation on the children’s diagnoses (Table 3). Aspects such as the deletion size could be useful in future qualitative research. Some authors, such as Cammarata-Scalisi et al. [45], reported that it is directly correlated to the disability [45] and it may influence the parental experience. In contrast, Ziats et al. [47] suggested that there is a general lack of correlation between terminal deletion size and phenotype. Finally, it is important to consider the perspectives of the siblings for a more comprehensive analysis of the impact of PMS.

## 5. Conclusions

The parents of children with PMS undergo a process of adaptation and acceptance of their child’s rare disease, characterized by flexibility and living day by day, setting short-term goals, and solving multiple daily difficulties. Our results show how the parents’ expectations of building “their ideal family” were shattered by the reality of the disease and how their future and that of their child is full of uncertainty. However, despite the pain, their child is a source of joy and happiness that has helped them to have a broader and deeper perspective on life.

Our results can be used by health professionals to help and support parents. Moreover, programs integrating health and social interventions seems to be more appropriated according to the experience of parents with children with PMS.

## Figures and Tables

**Table 1 children-10-00073-t001:** In-depth interviews’ question guide.

Areas to Be Investigated	Questions
Symptoms	When did you become aware of the onset of symptoms? Which are the most relevant to you? What do these symptoms mean to you?
Diagnosis	How was the process to get a diagnosis? What was the most relevant part of this process?
Treatment and care	What treatment is your child currently receiving? What is the most relevant aspect to you? Does it affect or limit your daily life? How? Are there any side effects? What are your expectations for the treatment? Do you perceive any barriers and facilitators to go through the recommended treatment? Do you use any coping strategies to deal with the symptoms of the disease? Can you describe the caregiving to your child? What are the barriers and/or facilitators of your child caregiving?
Illness	What is your experience of living with Phelan McDermid Syndrome? Which is the most relevant aspect for you? How does it affect your routine? What are your expectations about the syndrome and its evolution?
Impact on parents	How do you experience your child’s illness? What feelings and/or thoughts do you have? What is most relevant?

**Table 2 children-10-00073-t002:** Criteria used to control trustworthiness.

Criteria	Techniques Performed
Credibility	Investigator triangulation: each interview was analyzed by two researchers. Team meetings were performed. Triangulation of data collection methods: semistructured interviews and researcher field notes were used. Member-checking. All participants were asked to review their audio and/or video records to confirm their perspective and no comments were added.
Transferability	In-depth and detailed descriptions of the research process were performed.
Dependability	Audits were applied by an external researcher on the research protocol (the methods applied and study design). Also, another external researcher checked the codification process.
Confirmability	Investigator triangulation, member-checking, and data collection triangulation. Researcher reflexivity was encouraged via the performance of reflexive reports.

**Table 3 children-10-00073-t003:** Sociodemographic characteristics of each participant and clinical features their children.

Code	Sex	Age	Marital Status	Age at Diagnosis (Years)	Age of Child with PMS (Years)	Diagnosis
E1	Male	40	Married	3	12	Deletion
E2	Female	46	Married	3	12	Deletion
E3	Female	46	Married	3	16	Deletion
E4	Female	39	Married	1.5	8	Deletion
E5	Female	49	Divorced	0.9	20	Deletion
E6	Female	35	Married	0.5	3	Deletion
E7	Male	38	Married	0.5	3	Deletion
E8	Female	54	Married	1	22	Deletion
E9	Female	38	Married	0.3	4	Deletion
E10	Female	39	Married	9	11	SHANK3 gene mutation
E11	Female	42	Divorced	3	9	Deletion
E12	Female	62	Divorced	34	36	Deletion
E13	Female	52	Married	16	18	Deletion
E14	Female	46	Married	6	9	SHANK3 gene mutation
E15	Female	38	Married	3	6	Deletion
E16	Male	39	Married	3	6	Deletion
E17	Female	34	Married	7	13	Deletion
E18	Male	42	Married	3	6	Deletion
E19	Female	44	Divorced	1.5	3	Deletion
E20	Female	38	Married	1	2	Deletion
E21	Female	60	Married	33	35	Deletion
E22	Male	40	Married	1.5	8	Deletion
E23	Male	52	Married	16	18	Deletion
E24	Male	38	Married	3	5	Deletion
E25	Female	38	Married	3	5	Deletion
E26	Female	46	Married	7	10	Deletion
E27	Female	48	Divorced	39	40	Deletion
E28	Female	45	Divorced	9	10	Deletion
E29	Female	40	Married	3	5	Deletion
E30	Male	43	Married	3	5	Deletion
E31	Female	36	Married	3.5	5	Deletion
E32	Male	41	Divorced	4	8	Deletion

## Data Availability

The database can be accessed on request from the corresponding author.

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
