# Peer review of "“Your Life Turns Upside Down”: A Qualitative Study of the Experiences of Parents with Children Diagnosed with Phelan-McDermid Syndrome"

_children, 2022, doi:10.3390/children10010073_

Round 1

Reviewer 1 Report

“Your life turns upside down”:  A qualitative study of the experiences of parents with children diagnosed with Phelan-McDermid syndrome”

BRIEF SUMMARY:

This qualitative exploratory study recruits 32 parents of children diagnosed with Phelan-McDermid syndrome (PMS), who are present at an Association for Families with PMS meeting.  Over a four-month period, the researchers interview these parents in a private video chat room, using a semi-structured question guide in order to explore each parent’s experience having a child with this rare disorder.  The results of these audio and visual recordings, along with researchers’ field notes, were analyzed using an inductive thematic analysis and revealed the following themes:  Understanding and Accepting the Disease, Living Day by Day, Expectations versus Reality, and Suffering and Happiness.  The authors cite examples of parents’ comments that are relevant to each theme, also noting their consistency with much of the existing literature concerning the experiences of parents with children afflicted by rare disease.  This article adds to the existing literature, however, as the authors claim that there are no previous qualitative studies describing the experience of parents with children diagnosed with PMS.  This study should be considered for publication with major revisions offered below.

 ABSTRACT:

Lines 42-45:  Lines 42-45 are confusing and need to be re-ordered as suggested:

To clarify, the authors could begin the abstract with a statement concerning the experiences found in the parents of children with other (not PMS) rare diseases.  These descriptions can then be cited.   The next sentence of the abstract (Lines 44-45) would then follow logically, as the current article has a clear purpose of exploring the experience of parents with children diagnosed with PMS”.  The later statement (Lines 73-74) stating the novelty of the current study’s objective is no longer contradictory. 

INTRODUCTION:

Line 66:  This statement could be softened to say, “likely underdiagnosed” or “PMS is thought to be underdiagnosed and its real incidence is unknown due to…”.

Lines 75-76:  “The purpose of this study… and daily care of children” needs to be clarified.  What does “meaning of disease” mean?  “Expectations” in relation to what exactly?   The “daily care of children” might be more meaningful with a few extra words, such as, “the challenges and rewards of daily care”.

MATERIALS AND METHODS:

Line 99:  Remove the word, “and”.

Table 1:  The interviewer might benefit from organizing the “Areas to Be Investigated” in such a way that they flow with a natural conversation.  For example, this might start with the recognition of “Symptoms”, which then leads to discussions of the journey toward “Diagnosis” and the resulting “Treatment and Care” obtained post-diagnosis.  Discussions related to the “Impact on Parents” and “Illness” can logically follow these previous categories.

Table 1:  Authors can consider the inclusion of questions that more directly attempt to extract data that will be useful for other parents.  For example, when interviewing more experienced parents of older children, questions might include:

·       What do you wish you’d been told at the time you found out about the diagnosis?  At any other earlier stages?

·       Has more time with the diagnosis changed your expectations or outlook on the disease?

·       What have been the best resources for you? 

The authors might also consider the benefits of correlating particular questions to parents of particular age groups (i.e. birth-3 years, 4-8 years, etc).  This could allow questions by the researcher to be tailored to the issues of each respective age group.  For example, parents could be queried about increases/decreases in particular behaviors or other symptoms at various ages or stages of development.  This data can then be used to alert and train/prepare the parents to address these issues successfully. 

RESULTS:

Table 3:  Needs revision: 

·       Could the headings of column 7 and 8 be reversed?  The data in these columns needs units of measurement, for clarification.

·       The columns, “Number of Children” and “Children with PMS” seem irrelevant.

·       The “deletions” in the “Diagnosis” column should be specified.

·       Five line pairs with almost identical data (i.e. E1-E2, E6-E7, E15-E16), are found in the table.  Is the reader to infer that each of these pairs of lines represents a couple who have the same child with PMS?  This could be clarified.

DISCUSSION:

Authors discuss the emergent themes found in the interview and field notes.  Similar findings in the literature concerning other rare diseases, are referenced.

Author Response

RESPONSE LETTER children-2092673

Manuscript ID children-2092673

Entitled: “Your life turns upside down”: A qualitative study of the experiences of parents with children diagnosed with Phelan-McDermid syndrome.”

Journal: Children.

Special issue: Advances in Rehabilitation of Children with Disabilities.

We would like to thank the Editors and the Reviewers for their careful consideration of our manuscript. We would also like to thank the Reviewers for their suggestions, which we believe have enhanced the quality of the manuscript. We have highlighted (in yellow) all the changes we have made throughout the text. Below, please find a detailed list of how we have addressed each comment.

Review Report (Reviewer 1)

Comments and Suggestions for Authors

BRIEF SUMMARY:

This qualitative exploratory study recruits 32 parents of children diagnosed with Phelan-McDermid syndrome (PMS), who are present at an Association for Families with PMS meeting.  Over a four-month period, the researchers interview these parents in a private video chat room, using a semi-structured question guide in order to explore each parent’s experience having a child with this rare disorder.  The results of these audio and visual recordings, along with researchers’ field notes, were analyzed using an inductive thematic analysis and revealed the following themes:  Understanding and Accepting the Disease, Living Day by Day, Expectations versus Reality, and Suffering and Happiness.  The authors cite examples of parents’ comments that are relevant to each theme, also noting their consistency with much of the existing literature concerning the experiences of parents with children afflicted by rare disease.  This article adds to the existing literature, however, as the authors claim that there are no previous qualitative studies describing the experience of parents with children diagnosed with PMS.  This study should be considered for publication with major revisions offered below.

RESPONSE:

Thank you for the opportunity to respond to your suggestions.

ABSTRACT:

Lines 42-45:  Lines 42-45 are confusing and need to be re-ordered as suggested:

To clarify, the authors could begin the abstract with a statement concerning the experiences found in the parents of children with other (not PMS) rare diseases. These descriptions can then be cited. The next sentence of the abstract (Lines 44-45) would then follow logically, as the current article has a clear purpose of exploring the experience of parents with children diagnosed with PMS”.  The later statement (Lines 73-74) stating the novelty of the current study’s objective is no longer contradictory. 

RESPONSE:

Thank you for your suggestions. We have followed the reviewer’s suggestions. We now include new information regarding parents’ experiences. We were unable to include extensive information in order to comply with the journal's rules regarding the word count of the abstract.

Furthermore, citations of references have not been included in the "abstract" section in accordance with the journal's standards.

Please refer to the edited text below:

Abstract: (1) Background: Parents of children with rare diseases experience great uncertainty and employ different strategies to care for their children and cope with the disease. The purpose of this study was to explore the experience of parents with children diagnosed with Phelan-McDermid Syndrome (PMS); (2)…

INTRODUCTION:

Line 66:  This statement could be softened to say, “likely underdiagnosed” or “PMS is thought to be underdiagnosed and its real incidence is unknown due to…”.

RESPONSE:

Edited as suggested by the reviewer.

Lines 75-76:  “The purpose of this study… and daily care of children” needs to be clarified.  What does “meaning of disease” mean?  “Expectations” in relation to what exactly?   The “daily care of children” might be more meaningful with a few extra words, such as, “the challenges and rewards of daily care”.

RESPONSE:

We have followed the reviewer’s suggestions.

We have included new information to clarify the objective of the study.

See edited text below:

The purpose of this study was therefore to explore the experience of parents with children diagnosed with PMS regarding how they integrate the disease into their daily life, expectations regarding the disease and treatment, and the challenges and rewards involved in the daily care of their children.

MATERIALS AND METHODS:

Line 99:  Remove the word, “and”.

RESPONSE:

Edited as suggested by the reviewer.

Furthermore, that sentence has been modified in order to expand on the information required by reviewer 3.

During the interviews, the investigators used prompts or probes: a) to encourage patients to elaborate further (Can you tell me more about that?), b) to encourage participants to keep talking (Have you experienced the same thing again?), c) to resolve doubts (paraphrase something the patient has said), and d) to show that they have the researcher's full attention (That's very interesting, please tell me more).

Table 1:  The interviewer might benefit from organizing the “Areas to Be Investigated” in such a way that they flow with a natural conversation.  For example, this might start with the recognition of “Symptoms”, which then leads to discussions of the journey toward “Diagnosis” and the resulting “Treatment and Care” obtained post-diagnosis.  Discussions related to the “Impact on Parents” and “Illness” can logically follow these previous categories.

RESPONSE:

We have followed the reviewer’s suggestions.

Table 1:  Authors can consider the inclusion of questions that more directly attempt to extract data that will be useful for other parents.  For example, when interviewing more experienced parents of older children, questions might include:

  • What do you wish you’d been told at the time you found out about the diagnosis?  At any other earlier stages?
  • Has more time with the diagnosis changed your expectations or outlook on the disease?
  • What have been the best resources for you? 

RESPONSE:

The authors agree with the reviewer, however, because the study has already been conducted and completed, these questions cannot be included. However, we believe that it is essential that they be taken into account for future studies. For this reason, we have included the reviewer's suggestions in the discussion section (limitations). See the new text below:

In addition, future studies could include questions that would be useful for other parents with children with other RD. For example, they could include: a) What do you wish you’d been told at the time you found out about the diagnosis? At any other earlier stages?; b) Has more time since the diagnosis changed your expectations or outlook on the disease?; and c) What are the best resources you have encountered?

The authors might also consider the benefits of correlating particular questions to parents of particular age groups (i.e. birth-3 years, 4-8 years, etc).  This could allow questions by the researcher to be tailored to the issues of each respective age group.  For example, parents could be queried about increases/decreases in particular behaviors or other symptoms at various ages or stages of development. This data can then be used to alert and train/prepare the parents to address these issues successfully. 

RESPONSE:

The authors agree with the reviewer. However, during the study we did not apply this distribution of the questions by age range or the children's stage of development. However, we agree with the reviewer that this is an important aspect. Additional information is included in the limitations section of the manuscript.

We have included the following text:

Another limitation is that the questions asked to the parents have not been selected in accordance with the child's age (e.g., birth-3 years, 4-8 years, etc.), nor have they been asked at specific moments in time, such as immediately after diagnosis. This could enable inquiries into the parents' perspective on aspects such as the increase or decrease of certain behaviors or other symptoms at different developmental stages. In addition, future studies could include questions that would be useful for other parents with children with other RD. For example, they could include: a) What do you wish you’d been told at the time you found out about the diagnosis? At any other earlier stages?; b) Has more time since the diagnosis changed your expectations or outlook on the disease?; and c) What are the best resources you have encountered?

RESULTS:

Table 3:  Needs revision: 

  • Could the headings of column 7 and 8 be reversed?  The data in these columns needs units of measurement, for clarification.

RESPONSE: We have followed the reviewer’s suggestions. There was an error and it has been corrected by changing the information in column 7 and 8. The unit of measurement (years) is now included.

  • The columns, “Number of Children” and “Children with PMS” seem irrelevant.

RESPONSE: Edited as suggested.

  • The “deletions” in the “Diagnosis” column should be specified.

RESPONSE: We agree with the reviewer. However, the present study did not include information regarding deletion size, only the presence of deletion and SHANK3 gene mutation (see Table 3. Demographic and clinical features). Furthermore, the authors believe that deletion size would have influenced the parents’ experience. For this reason, we have included information about this in the limitations section.

See the following text below:

Moreover, the present study solely reported the presence of deletion and SHANK3 gene mutation on the children's diagnoses (Table 3). Additional details about deletion size would be useful in future qualitative research because Cammarata-Scalisi et al [45] reported that larger deletions correlate to more severe disability and that may have influenced the parental experience. In contrast, Ziats et al. [48] suggested that there is a general lack of correlation between terminal deletion size and phenotype. Finally, it is important to consider the perspectives of the siblings for a more comprehensive analysis of the impact of PMS.

We have included new references:

45.Cammarata-Scalisi, F.; Callea, M.; Martinelli, D.; Willoughby, C.E.; Tadich, A.C.; Araya Castillo, M.; Lacruz-Rengel, M.A.; Medina, M.; Grimaldi, P.; Bertini, E.; Nevado, J. Clinical and Genetic Aspects of Phelan-McDermid Syndrome: An Interdisciplinary Approach to Management. Genes (Basel). 2022, 13(3). https://doi.org/10.3390/genes13030504

48.Ziats C.A.; Grosvenor, L.P.; Sarasua, S.M.; Thurm, A.E.; Swedo, S.E.; Mahfouz, A.; Rennert, O.M.; Ziats, M.N. Functional genomics analysis of Phelan-McDermid syndrome 22q13 region during human neurodevelopment. PLoS One. 2019, 14(3):e0213921. doi: 10.1371/journal.pone.0213921.

  • Five line pairs with almost identical data (i.e. E1-E2, E6-E7, E15-E16), are found in the table.  Is the reader to infer that each of these pairs of lines represents a couple who have the same child with PMS?  This could be clarified.

RESPONSE: We have followed the reviewer’s suggestions.

We agree with the reviewer that this should be noted in the results section. We included in the results section:

The participants included in the present study were one of the parents, except in cases E1-E2, E6-E7, and E15-E16, where both spouses from the same family agreed to participate.

DISCUSSION:

Authors discuss the emergent themes found in the interview and field notes.  Similar findings in the literature concerning other rare diseases, are referenced.

RESPONSE: Thank you for your comment. 

We hope that you are satisfied with the revision and that the manuscript is now suitable for publication in Children.

Sincerely,

The Authors

Reviewer 2 Report

The topic of the article is important, the approach used is interesting - the search for themes in the way of perception of the disease by the parents of a child with a serious medical problem (PMS).

 The article is well constructed, well presented for analysis.

The literature review as well as the discussion lacks the contradictions in the literature on the issue. On the one hand, it is important to ascertain the reaction of parents and their experiences of a serious medical problem with their children. On the other hand, what makes the behavior of families of parents with this specific syndrome different from the behavior and reactions of parents, for example, of children with autism or mental retardation. With them, will we find other themes /experiences/ in the perception of the disease by the parents or will they be the same or similar? I did not find such a discussion in the presented article. There is no answer to other questions that arise in the analysis of the presented work.

 For example:

What is the impact of the disease on family integrity compared to that of the general population?

What are your thoughts on the dynamics of these themes. Do you think they will change and flow into each other over time and end with call it an ultimate /outline/ theme /pattern/?

In this sense, whether the evaluation of the family and its theme of perception of the disease does not depend on the time of the evaluation, i.e. time since diagnosis?

Aren't the themes you identified behavioral patterns typical of families with severe health problems for children in general, not just those with this medical problem?

I would be glad if the answers to these questions could find a place in your work

Reviewer

Author Response

RESPONSE LETTER children-2092673

Manuscript ID children-2092673

Entitled: “Your life turns upside down”: A qualitative study of the experiences of parents with children diagnosed with Phelan-McDermid syndrome.”

 Journal: Children.

Special issue: Advances in Rehabilitation of Children with Disabilities.

We would like to thank the Editors and the Reviewers for their careful consideration of our manuscript. We would also like to thank the Reviewers for their suggestions, which we believe have enhanced the quality of the manuscript. We have highlighted (in yellow) all the changes we have made throughout the text. Below, please find a detailed list of how we have addressed each comment.

Review Report (Reviewer 2)

Comments and Suggestions for Authors

The topic of the article is important, the approach used is interesting - the search for themes in the way of perception of the disease by the parents of a child with a serious medical problem (PMS).

RESPONSE:

Thank you for your comment. 

The article is well constructed, well presented for analysis.

RESPONSE:

Thank you for your comment. 

The literature review as well as the discussion lacks the contradictions in the literature on the issue. On the one hand, it is important to ascertain the reaction of parents and their experiences of a serious medical problem with their children. On the other hand, what makes the behavior of families of parents with this specific syndrome different from the behavior and reactions of parents, for example, of children with autism or mental retardation. With them, will we find other themes /experiences/ in the perception of the disease by the parents or will they be the same or similar? I did not find such a discussion in the presented article.

RESPONSE:

We have pointed out in the discussion how the physical changes that are characteristic of PMS may explain the behavior. We have described the case of parents with children with Down syndrome, where children have more marked physical characteristics, have similar behaviors and experiences to parents with PMS.  Specifically in the discussion, section 4.1. Understanding and accepting the disease, the following text has been inserted:

Children with PMS may present with nonspecific dysmorphic features such as large fingers, long eyelashes, and large ears, although these features are not characteristic of the syndrome [1,26]. However, in Down syndrome, where the presence of physical traits is more marked, parents report the same lack of understanding. They are confronted with comments from people on a daily basis as a result of the child’s behavior [27].

Further information has been provided:

Regardless of whether or not physical features are present, Currie & Szabo [24] report that in the face of signs of RD and when complications become visible to others, they feel a stigma from the rest of society that takes an emotional toll on them.

The final paragraph now reads:

Children with PMS may present with nonspecific dysmorphic features such as large fingers, long eyelashes, and large ears, although these features are not characteristic of the syndrome [1,26]. However, in Down syndrome, where the presence of physical traits is more marked, parents report the same lack of understanding. They are confronted with comments from people on a daily basis as a result of the child’s behavior [27]. Regardless of whether or not physical features are present, Currie & Szabo [24] report that in the face of signs of RD and when complications become visible to others, they feel a stigma from the rest of society that takes an emotional toll on them.

We have included a new reference:

24.Currie, G.; Szabo, J. Social isolation and exclusion: the parents’ experience of caring for children with rare neurodevel-opmental disorders. Int J Qual Stud Health Well-Being. 2020, 15(1):1725362. https://doi.org/10.1080/17482631.2020.1725362

There is no answer to other questions that arise in the analysis of the presented work.

For example:

What is the impact of the disease on family integrity compared to that of the general population?

What are your thoughts on the dynamics of these themes. Do you think they will change and flow into each other over time and end with call it an ultimate /outline/ theme /pattern/?

RESPONSE:

In relation to the first question, the authors, with the present study based on a qualitative design, cannot establish the impact of the disease (PMS) in families and its comparison with other diseases at the population level, because it is not an ecological or quantitative observational study. Likewise, we cannot establish inferences about its evolution and progression in relation to the general population, because this was not the objective of the present study. Regarding the change over time in the experience of parents with children with PMS, in terms of accepting and understanding the disease, living day to day, real expectations and the perspective of their child's future, the authors believe that there will be changes in the parents' experience as they better understand the disease, know what care and needs their children have, learn to manage the multiple symptoms of PMS, and their expectations will adapt to the specific reality of each family. Regarding the future, this is a sensitive issue, since parents show little confidence that a drug or curative treatment for the disease will be available for their children.

We have included the following information in the discussion section:

Comparing our results with other RD, a change in the experience of parents with children with RD is described to the extent that their understanding of the disease improves [18], they become aware of their children's needs and care requirements [24], they learn to manage the multiple symptoms of RD, and their expectations are adapted to the specific reality of each family [24,43]. Nonetheless, the future is a sensitive issue, because parents express limited confidence that a drug or curative treatment for the disease will be available for their children [43,44].

We have included new references:

24.Currie, G.; Szabo, J. Social isolation and exclusion: the parents’ experience of caring for children with rare neurodevel-opmental disorders. Int J Qual Stud Health Well-Being. 2020, 15(1):1725362. https://doi.org/10.1080/17482631.2020.1725362

43.Cheung, M.; Rylands, A.J.; Williams, A.; Bailey, K.; Bubbear, J. Patient-Reported Complications, Symptoms, and Experi-ences of Living With X-Linked Hypophosphatemia Across the Life-Course. J Endocr Soc. 2021, 5(8):bvab070. https://doi.org/10.1210/jendso/bvab070

44.Reeder, J.; Morris, J. Managing the uncertainty associated with being a parent of a child with a long-term disability. Child Care Health Dev. 2021, 47(6):816-824. https://doi.org/10.1111/cch.12889

In this sense, whether the evaluation of the family and its theme of perception of the disease does not depend on the time of the evaluation, i.e. time since diagnosis?

RESPONSE:

The authors agree with the reviewer. We believe that this observation should be included in the limitations section.

The following text has been included:

Another limitation is that the questions asked to the parents have not been selected in accordance with the child's age (e.g., birth-3 years, 4-8 years, etc.), nor have they been asked at specific moments in time, such as immediately after diagnosis. This could enable inquiries into the parents' perspective on aspects such as the increase or decrease of certain behaviors or other symptoms at different developmental stages. In addition, future studies could include questions that would be useful for other parents with children with other RD. For example, they could include: a) What do you wish you’d been told at the time you found out about the diagnosis? At any other earlier stages?; b) Has more time since the diagnosis changed your expectations or outlook on the disease?; and c) What are the best resources you have encountered?

Aren't the themes you identified behavioral patterns typical of families with severe health problems for children in general, not just those with this medical problem?

RESPONSE:

The authors do not believe that the patterns and/or themes identified are similar to those of families with children with severe illnesses or health problems, because parents with children with PMS and/or RD face greater difficulties and challenges. Among these difficulties are the fact that PMS and/or RD have no clear curative treatment, the pathophysiological mechanisms of many RD are not known with certainty, and there is no mass knowledge of RD or PMS. Compared to other pediatric diseases such as pneumonia or bronchiolitis or a hematologic cancer such as lymphoma. In addition, since PMS and RD are minority diseases, health resources do not have sufficient coverage to cover all the needs of these children and their families.

The following new text has been included:

The authors of the present study believe that a key difference between parents with children with PMS and/or RD and parents with children with severe health problems is that families with PMS and/or RD are more vulnerable because: a) PMS has no known curative treatment [7,45], b) there are problems related to its diagnosis that result in a diagnostic delay [46], and c) there are not enough resources in the health system for the study of PMD and/or RD because they are a minority [45-47].

We have included new references:

  1. Cammarata-Scalisi, F.; Callea, M.; Martinelli, D.; Willoughby, C.E.; Tadich, A.C.; Araya Castillo, M.; Lacruz-Rengel, M.A.; Medina, M.; Grimaldi, P.; Bertini, E.; Nevado, J. Clinical and Genetic Aspects of Phelan-McDermid Syndrome: An Inter-disciplinary Approach to Management. Genes (Basel). 2022, 13(3). https://doi.org/10.3390/genes13030504
  2. Phelan, K.; Rogers, R.C.; Boccuto, L. Phelan-McDermid Syndrome. En Adam, M.P.; Mirzaa, G.M.; Pagon, R.A.; Wallace, S.E.; Bean, L.J.; Gripp, K.W.; Amemiya, A. (Eds.), GeneReviews®. 2018, University of Washington, Seattle. http://www.ncbi.nlm.nih.gov/books/NBK1198/
  3. Bassell, J.; Srivastava, S.; Prohl, A.K.; Scherrer, B.; Kapur, K.; Filip-Dhima, R.; Berry-Kravis, E.; Soorya, L.; Thurm, A.; Powell, C.M.; Bernstein, J.A.; Buxbaum, J.D.; Kolevzon, A.; Warfield, S.K.; Sahin, M. Diffusion Tensor Imaging Abnor-malities in the Uncinate Fasciculus and Inferior Longitudinal Fasciculus in Phelan-McDermid Syndrome. Pediatr Neurol. 2020, 106:24-31. https://doi.org/10.1016/j.pediatrneurol.2020.01.006

I would be glad if the answers to these questions could find a place in your work.

RESPONSE:

Thank you for the opportunity to respond to your suggestions.

We hope that you are satisfied with the revision and that the manuscript is now suitable for publication in Children.

Sincerely,

The Authors

Reviewer 3 Report

Dear authors,

Please find my comments to your manuscript in the document attached.

Kind regards

Author Response

RESPONSE LETTER children-2092673

Manuscript ID children-2092673

Entitled: “Your life turns upside down”: A qualitative study of the experiences of parents with children diagnosed with Phelan-McDermid syndrome.”

 Journal: Children.

Special issue: Advances in Rehabilitation of Children with Disabilities.

We would like to thank the Editors and the Reviewers for their careful consideration of our manuscript. We would also like to thank the Reviewers for their suggestions, which we believe have enhanced the quality of the manuscript. We have highlighted (in yellow) all the changes we have made throughout the text. Below, please find a detailed list of how we have addressed each comment.

Review report 3

I would like to congratulate the authors for the study they present.

RESPONSE:

Thank you for your comment. 

  1. Comments

My comments are organized below for your consideration. I hope my comments are useful for the study authors and editorial staff.

RESPONSE:

Thank you for the opportunity to respond to your suggestions.

#1 The following sentence must be deleted from de sub-heading 2.1. Study design: “Ethical approval was provided by the Local Ethical Committee of Universidad [BLINDED] (code: 81 0810202017820)”. I suggest the authors create the sub-heading 2.3 Ethical Considerations and transfer to there the sentence about the ethical approval. There authors should also add information about participants informed consent and about other confidentiality and data security issues.

RESPONSE: We have followed the reviewer’s suggestions.

See the new sub-heading and text below:

2.3 Ethical Considerations

Ethical approval was provided by the Local Ethical Committee of Universidad [BLINDED] (code: 0810202017820). Prior to data collection, participants were asked for permission and informed consent was obtained. The data obtained were anonymized and were not shared with anyone outside the research team.

#2 The authors must explain in a more clearly way the following statement: “During the interviews, researchers used prompts and to encourage the participant to provide further detail” (Line 99).

RESPONSE:

We agree with the reviewer.

The researchers used prompts or probes to stimulate participants to answer, provide more information, resolve doubts or clarify points. These probes were used throughout the interview.

This has been clarified in the data collection section:

During the interviews, the investigators used prompts or probes: a) to encourage patients to elaborate further (Can you tell me more about that?), b) to encourage participants to keep talking (Have you experienced the same thing again?), c) to resolve doubts (paraphrase something the patient has said), and d) to show that they have the researcher's full attention (That's very interesting, please tell me more).

#3 The authors must substitute the comma (“,”) for a point (“.”) in the number in the following sentence: “, a total of 3,205 min of interviews were recorded” (Line 108), as required by the authors guidelines of the Journal.

RESPONSE: We believe if we substitute the comma for a point this may lead to misunderstanding as it may seem like three minutes instead of three thousand minutes, considering that in English commas are normally used to separate numbers greater than 999. https://prowritingaid.com/grammar/1008091/When-should-I-use-a-comma-to-separate-numbers

 We were unable to find this requirement in the journal guidelines. To avoid this confusion we have removed the comma and also the period.

#4 In the sub-heading 2.4. Data analysis, the authors must mention the triangulation process, and also if they used or not any kind of software to do the content analysis of all the data collected (for example: NVivo or MAXQDA or other).

RESPONSE: We have followed the reviewer’s suggestions.

The triangulation process is described in section 2.6. Rigor. Within the techniques established for credibility control, the triangulation of the analysis is described in Table 2. It was included in Table 2:

Criteria

Techniques Performed and Application Procedures

Credibility

Investigator triangulation: each interview was analyzed by two researchers. Team meetings were performed in which the analyses were compared and categories and themes were identified.

For data analysis, Excel software was used to construct matrices and templates during the coding process.

We included the following in the analysis section:

The Excel program was used to organize and share the coding process.

#5 In the sub-heading 4.2. Living day by day, the authors don’t present discussion about the constant need that the parents have to explain the behavior of their children, and also about the constant stress and tension of the parents, and the authors mention those aspects in this part of the Results heading.

RESPONSE: We have followed the reviewer’s suggestions.

Concerning “the behavior of their children” we have included more information in the subheading 4.2.

We have expanded the discussion in this section of the discussion as follows:

The presence of behavioral problems and/or autistic traits in PMS appears most frequently between childhood and adolescence [5,7,30]. Nayeri et al [31] reflect the difficult situations experienced by parents due to the misunderstanding that people display towards the behaviors of their children with a RD. Thus, the generalized experience of parents of children with a diagnosis of RD, Down syndrome or ASD shows a generalized lack of understanding of others regarding their children's disruptive behaviors [24-27]. As a result, they report a constant struggle to avoid stigma and lack of understanding by the public. It involves mental and physical overexertion and exhaustion, as well as stress [24,32]. Bolbocean et al [33] and Vogels et al [34] state that families of people with PMS are at high risk of mental health problems such as depression, stress and/or anxiety associated with the caregiving role.

#6 In the sub-heading 4.3. Expectations versus reality, the authors don’t present discussion about the suffering that parents feel when comparing themselves to other families, and the authors mention those aspects in this part of the Results heading.

RESPONSE: We have followed the reviewer’s suggestions.

We included further information concerning sub-heading 4-3:

Sometimes parents compare their children with other healthy children, which is especially painful for them. However, parents report that, after doing so, they regret these comparisons, especially in the early stages of RD as it is a very hard process for them and causes great suffering [24].

#7 In the sub-heading 4.4. Suffering and happiness, the authors don’t present discussion about the about the feeling of guilt that parents feel about some of their thoughts, and also about the fact that they consider that having a child with PMS has made them better people.

RESPONSE: We have followed the reviewer’s suggestions.

The following text has been included in the discussion section:

Moreover, previous studies have shown the appearance of feelings of guilt and suffering in the day-to-day life of parents of children with RD. Awareness of the present and future limitations of children with a RD causes a situation that they define as difficult and painful for the parents [24,31,36,41,42]. However, living with a child with a RD shows them the positive and important things in life [24].

We hope that you are satisfied with the revision and that the manuscript is now suitable for publication in Children.

Sincerely,

The Authors

Round 2

Reviewer 2 Report

The authors have complied with a large part of the remarks made and in this form the article can be published.

Reviewer